# A Gene Co-Expression Network-Based Drug Repositioning Approach Identifies Candidates for Treatment of Hepatocellular Carcinoma

**DOI:** 10.3390/cancers14061573

**Published:** 2022-03-19

**Authors:** Meng Yuan, Koeun Shong, Xiangyu Li, Sajda Ashraf, Mengnan Shi, Woonghee Kim, Jens Nielsen, Hasan Turkez, Saeed Shoaie, Mathias Uhlen, Cheng Zhang, Adil Mardinoglu

**Affiliations:** 1Science for Life Laboratory, KTH—Royal Institute of Technology, SE-17165 Stockholm, Sweden; meng.yuan@scilifelab.se (M.Y.); ko.shong@scilifelab.se (K.S.); xiangyu.li@scilifelab.se (X.L.); mengnan.shi@scilifelab.se (M.S.); woonghee.kim@scilifelab.se (W.K.); saeed.shoaie@kcl.ac.uk (S.S.); mathias.uhlen@scilifelab.se (M.U.); 2Bash Biotech Inc., 600 West Broadway, Suite 700, San Diego, CA 92101, USA; 3Heka Lab, Camlik Mah. Hearty, Sk. No:4 Heka Human Plaza Umraniye, Istanbul 34774, Turkey; sajda.ashraf@yahoo.com; 4Department of Biology and Biological Engineering, Chalmers University of Technology, SE-41296 Gothenburg, Sweden; nielsenj@chalmers.se; 5BioInnovation Institute, DK-2200 Copenhagen, Denmark; 6Department of Medical Biology, Faculty of Medicine, Atatürk University, Erzurum 25240, Turkey; hturkez@atauni.edu.tr; 7Centre for Host-Microbiome Interactions, Faculty of Dentistry, Oral & Craniofacial Sciences, King’s College London, London SE1 9RT, UK; 8Key Laboratory of Advanced Drug Preparation Technologies, School of Pharmaceutical Sciences, Ministry of Education, Zhengzhou University, Zhengzhou 450001, China

**Keywords:** systems biology, co-expression network, survival analysis, drug repositioning, hepatocellular carcinoma (HCC)

## Abstract

**Simple Summary:**

Hepatocellular carcinoma (HCC) is the most common malignancy of liver cancer. However, treatment of HCC is still severely limited due to limitation of drug therapy. We aimed to screen more possible target genes and candidate drugs for HCC, exploring the possibility of drug treatments from systems biological perspective. We identified ten candidate target genes, which are hub genes in HCC co-expression networks, which also possess significant prognostic value in two independent HCC cohorts. The rationality of these target genes was well demonstrated through variety analyses of patient expression profiles. We then screened candidate drugs for target genes and finally identified withaferin-a and mitoxantrone as the candidate drug for HCC treatment. The drug effectiveness was validated in in vitro model and computational analysis, providing more evidence for our drug repositioning method and results.

**Abstract:**

Hepatocellular carcinoma (HCC) is a malignant liver cancer that continues to increase deaths worldwide owing to limited therapies and treatments. Computational drug repurposing is a promising strategy to discover potential indications of existing drugs. In this study, we present a systematic drug repositioning method based on comprehensive integration of molecular signatures in liver cancer tissue and cell lines. First, we identify robust prognostic genes and two gene co-expression modules enriched in unfavorable prognostic genes based on two independent HCC cohorts, which showed great consistency in functional and network topology. Then, we screen 10 genes as potential target genes for HCC on the bias of network topology analysis in these two modules. Further, we perform a drug repositioning method by integrating the shRNA and drug perturbation of liver cancer cell lines and identifying potential drugs for every target gene. Finally, we evaluate the effects of the candidate drugs through an in vitro model and observe that two identified drugs inhibited the protein levels of their corresponding target genes and cell migration, also showing great binding affinity in protein docking analysis. Our study demonstrates the usefulness and efficiency of network-based drug repositioning approach to discover potential drugs for cancer treatment and precision medicine approach.

## 1. Introduction

The most common histologic type of primary liver cancer is hepatocellular carcinoma (HCC), and it accounts for 75–85% of cases of liver cancer with continuously increasing incidences and mortality worldwide [1]. HCC is a heterogeneous complex disease due to multiple etiological factors, including hepatitis B virus (HBV), hepatitis C virus (HCV), diabetes, alcoholic fatty liver disease (AFLD) and non-alcoholic fatty liver disease (NAFLD) [2]. However, the risk factors for HCC depend on the difference in geographic location, making the treatment options for HCC different and complicated. For example, HBV infection is the predominant cause in Japan and the US. However, different from Japan, HCV-infection-associated liver cancer death rate in the US remains highest among certain age populations [3], which may indicate the difference in pathology and etiologic factors with different geographic locations.

Several treatment modalities are available, including radiofrequency ablation, locoregional therapy and liver transplantation [4]. Systemic treatment, such as tyrosine-multikinase inhibitor sorafenib, was also indicated to be a potential for advanced stages of HCC, but it can only improve overall survival by approximately 3 months [5]. Other tyrosine-multikinase inhibitors, on the other hand, were found to be unsafe or did not improve the overall survival of patients [6]. The absence of effective prevention or treatment methods have encouraged extensive efforts to develop new therapeutic strategies against HCC [7], to offer more treatment options and actually improve patients’ survival outcome.

The expanding volume of multi-omics data and increasing number of public medical databases have enabled an alternative computational drug discovery approach and have been successfully employed to develop therapies for multiple cancer types [8,9,10]. Compared to traditional drug development strategies, computational drug repositioning is especially appealing, since the screening of candidate drugs is rapid [11], and the safety and pharmacokinetic profiles of the drugs are well known, which hugely reduces time and money needed for the entire drug development [9]. Computational drug repositioning establishes interactions among drugs, genes, diseases and proteins based on open-access databases and identifies effective drug candidates, presuming they target the same proteins in the treatment of diseases. The Connectivity Map (CMap) is considered as the key resource behind various well-acknowledged drug repositioning studies [12]. The CMap, which is also known as LINCS-L1000, contains more than 300,000 transcriptomic profiles generated from compounds and gene perturbagens across 248 cell lines [13], in which 978 landmark genes were defined by examining 12,063 expression profiles from the genotype tissue expression (GTEx) RNA-seq dataset.

Systems-biology-based drug repositioning approach is favorable, since it enables the identification of the key driver genes involved in disease progression in different systematic background contents, such as those based on differentially expressed genes [14] or interactive/regulatory networks [15,16]. The target genes screened by other approaches may not be the driver genes of the disease, limiting their effects on the disease treatment. Previously, we employed personalized genome-scale metabolic models and predicted three target genes, which potentially inhibit tumor growth in HCC patients [17].

In this study, we adopted gene co-expression network (GCN) analysis to cover all protein-coding genes other than metabolic genes, as the latter only cover around 20–25% of human genome [18,19]. We applied GCN analysis to extract the target genes by their topology attributes and prognostic effects estimated by survival analysis. The GCN-based analysis and prognostic screening enabled us to retrieve potential driver genes, which are hub genes in the network and expected to play important roles in the development of HCC. Next, we presented a drug repositioning method to identify effective drugs based on gene perturbation profiles and co-expression network analysis (Figure 1).

The method was conducted following five different steps: (1) extract a robust signature gene set, whose expression level highly associated with patient survival outcome, (2) identify prognostic modules that are conserved in different HCC cohorts, as well as hub genes in these modules, (3) extract potential targets in HCC, (4) screen effective drugs for each target based on the shRNA and drug perturbation transcriptomic data from the LINCS data portal and, finally, (5) evaluate the efficacy of selected drugs in in vitro models.

## 2. Materials and Methods

### 2.1. Data and Preprocessing

The gene expression profiles (fragment per kilobase of exon per million mapped fragments, FPKM) for LIHC cohort was downloaded from The Cancer Genome Atlas (TCGA) [20]. We screened all samples of liver hepatocellular carcinoma (LIHC) and kept 365 donors with both RNA-seq data obtained from primary tumor solid tissue samples and well-recorded clinical information. The FPKM profiles of protein-coding genes were obtained from the Human Pathology Atlas [21]. The clinical information was collected from TCGA Pan-Cancer Clinical Data Resource (TCGA-CDR) [22] by retrieving cancer type of LIHC.

The gene expression profiles (FPKM) and clinical information of Japanese cohort with 229 donors was collected from the International Cancer Genome Consortium (ICGC) (http://icgc.org/, 4 June 2021). Here, we only kept patients with primary tumor solid tissue samples and clinical information. For donors with multiple tumor samples, we followed the rule below to prevent the expression data from being ambiguous; namely, we kept ones with highest percentage of cellularity. All genes were mapped by GRch37 with protein-coding area for data consistency.

Among the two cohorts, only genes with average FPKM no less than 1 were seen as expressed genes and reserved for the following analysis to reduce the noise inherent in measuring lowly expressed genes. The correlation of gene expression level between LIHC and LIRI-JP cohort was evaluated by Spearman correlation. Only genes expressed in both cohorts were included and shown in Figure 2A. The analysis was performed on RStudio (R version 4.1.0).

### 2.2. Survival Analysis

Both the univariable Cox regression model and the Kaplan–Meier (KM) survival analysis were performed to evaluate the association between patients’ transcriptomic profiles and clinical survival outcomes, with time from enrolment to overall death (survival time less than 0 day was excluded). For univariable Cox analysis, we calculated the hazard ratio of each gene to show the association between gene expression and patients’ days alive. Genes with adjusted p value of less than 0.05 were kept as significant genes related with survival outcomes. For analysis specifically, patients were divided into subgroups with high and low expression levels of each gene to examine the prognostic effects of expression level. To choose the best FPKM cut-offs for grouping, all FPKM values from the 20th to 80th percentiles were used to group the patients into two groups. Significant differences in the survival outcomes of the groups were examined by log-rank tests. The value yields with the lowest p value was selected. Genes with False Discovery Rate (FDR) adjusted by Benjamini–Hochberg (BH) method of less than 0.05 were extracted as prognostic genes. In addition, if the group of patients with high expression of a selected prognostic gene had a higher observed survival time than the expected event, it was a favorable prognostic gene; otherwise, it was an unfavorable prognostic gene. All analyses were applied on R with package “survival”.

Genes with FDR of less than 0.05 were extracted on the bias of both survival analyses, and those with same prognostic implication (both favorable or unfavorable) in both Cox analysis and KM analysis among two cohorts were extracted as signature prognostic genes (SPGs) and adopted for further analysis.

### 2.3. Functional Enrichment Analysis

The R package “clusterProfiler” was implemented to find the enriched biological progress in SPGs. The gene ontology (GO) functional enrichment analysis was performed by the enrichGO function, and significantly enriched terms (FDR < 0.05, adjusted by BH method) were simplified by the dropGO function (level = 5) to eliminate redundant terms.

### 2.4. Co-Expression Analysis and Module Identification

Spearman correlation coefficients of gene expression level (between any two genes) were calculated to estimate gene co-expression association, and only the correlations ranked higher than 0.5% among all gene pairs were reserved to construct the GCN. Walktrap algorithm, which is able to identify densely connected modules based on structural similarity, was performed to identify gene modules with high transitivity in GCN. Only modules consisting of more than 20 nodes and with connectivity coefficients larger than 0.5 were extracted for further analysis. Next, we performed concordance analysis to identify the modules’ prognostic attributes. A module would be favorable if it was significantly overlapped with favorable SPGs (hypergeometric test, *p* < 0.05) or unfavorable in the same condition with unfavorable SPGs. Modules with no statistic overlapping with SPGs were then considered as non-prognostic modules. Then, we identified highly overlapped prognostic module pairs from the LIHC and LIRI-JP cohorts. Two prognostic modules (one from LIHC cohort, the other from LIRI-JP cohort) significantly overlapped (hypergeometric test) with FDR (adjusted by BH method) of less than 0.05 and were reserved as the most potential gene module pair in HCC, as they showed similar network gene component in two different cohorts. All analysis was performed on RStudio.

### 2.5. Identification of HCC Target Genes

To identify the influential genes of prognostic modules in GCN, topology analysis was performed to prioritize the hub genes with general influence over the prognostic module networks. Among the significantly overlapped prognostic module pair, three network topological parameters (degree, betweenness and closeness) of all involved genes were calculated, and genes were sorted in decreasing order in each module. Taking degree as an example, we extracted the genes with degree value ranking higher than 10 percent in each module. The hub genes shared by both two modules were then identified as degree-hub genes for this module pair. Similar steps were used to obtain betweenness-hub genes and closeness-hub genes. We then combined all three shared hub gene lists and regarded them as the most potential candidate genes for HCC drug targets.

The essential scores of genes in 16 primary HCC cell lines were downloaded from Cancer Dependency Map (DepMap) portal (https://depmap.org/portal/, 28 June 2021), which are estimated based on the CRISPR-Cas9 essentiality screens. A lower gene score indicates this gene possesses stronger effects on the tumor cell proliferation after CRISPR-Cas9 knockout of this gene. In most cases, a gene with essential score less than −0.5 means that the cell will grow slower when the gene is knocked out [23]. Here, potential target genes were selected by a much stricter criteria. Only genes with essential score less than −1.5 were reserved as most potential target genes for HCC.

### 2.6. Expression Differences of Target Genes

The mRNA expression profiles of genes in tumor and adjacent normal tissues were downloaded from the Toil [24] and processed to TPM by kallisto [25] for fair comparison. Here, we only included 50 patients diagnosed with HCC, with both tumor and adjacent normal tissue samples. The average TPM of tumor samples represents the average gene expression level in tumor tissue. Similarly, the average TPM of normal tissues represents the average expression level in normal tissue.

The immunohistochemistry (IHC) staining images of normal and tumor liver cells were downloaded from the Human Protein Atlas (HPA) (https://www.proteinatlas.org/, 14 September 2021). The generation of IHC and identification of staining intensity was described in their previous research [21]. For each target gene, we only selected the images generated by the same antibody for fair comparison.

### 2.7. The scRNA-Seq Data Processing

Single cell RNA-seq data of HCC were retrieved from open data source GSE166635 [26]. The raw counting data were downloaded, and the sample HCC1, which contains the greatest number of cells, was selected for further analysis to avoid batch effect. A standard pipeline of data analysis was performed using the scRNA-seq analysis package ScanPy (version 1.7.2) in Python 3.8.5. In the pipeline, cells that contained fewer than 200 genes and genes, which expressed in fewer than 1000 cells, were considered as invalid and removed from the data. Consequentially, 9868 genes across 14,258 cells were normalized from the cell counts into a total count per cell of 1.0E4 and then scaled by natural logarithm. Next, 2278 highly variable genes were detected by t-test, and Louvain clustering function was then applied to cluster valid cells, which bunched in 15 clusters. To annotate these clusters, pseudo-bulk data for each cluster were calculated by adding up each gene of all cells in corresponding cluster and then normalized to transcripts per million protein-coding genes (pTPM). Liver cell type annotation in HPA single cell atlas was downloaded as reference [27,28]. The Spearman correlation coefficients were then calculated across pseudo-bulk pTPM data of HPA liver and our liver data for each cluster, and the best matched cell type was picked up.

### 2.8. Drug Repositioning for HCC

The transcriptional response profiles of shRNA knockdown and chemical perturbation for HepG2, a liver carcinoma cell line, were downloaded from CMap LINCS 2020 (level 5 data). The level 5 data provide robust moderated z-score (differential expression values) of genes by at least 3-times-replicated biological experiments. Hence, we applied a drug repositioning method based on our previous research [29], integrating the shRNA knockout and compound perturbation profiles, with the hypothesis that a drug has a repressed effect on the expression of a target gene in tumor cells if drug treatment leads to a similar dysregulation of gene expression induced by shRNA knockout in cell lines. Altogether, we extracted 18,157 chemical compounds relative to 5284 drugs across multiple doses and time points, as well as 30 shRNAs among 7 target genes.

To identify promising candidate drugs for the target genes, we extracted signature for each target gene based on the gene expression changes after shRNA from CMap and matched the gene signature to all drug signatures available in CMap. The top 3 drugs whose signatures showed highest Spearman correlation with each of the target genes were considered as most efficient and selected as drug candidates. Detailed information is described in the Results section.

### 2.9. In Vitro Validation

#### 2.9.1. Cells Culture

Human HCC cell line HepG2 was purchased from genome engineering company Synthego. The cells were cultured in RPMI 1640 (R2405, Sigma-Aldrich, St. Louis, MO, USA) medium with 10% fetal bovine serum (FBS) and 1% penicillin/streptomycin. The cells were incubated in a humidified incubator (at 37 °C, 5% CO2)

#### 2.9.2. Drug Treatment

The drugs, Withaferin-a (WFA), Mitoxantrone (MTX), AZD-6482, CYT-387, Sulforaphane and Resveratrol were purchased from Selleckchem (Selleckchem, Houston, TX, USA). These drugs were dissolved in DMSO. The cells were seeded in a 96-well plate at 100 × 103 cells per well and 1 × 106 cells per well in a 6-well plate. When cell confluence became 90–95% of the well, drugs were added to the wells at a proper concentration: WFA (4 uM), MTX (10 uM), AZD-6482 (3.33 uM), CYT-387 (0.37 uM), Sulforaphane (2.5 uM) and Resveratrol (3.33 uM) and treated for 24 h.

#### 2.9.3. Western Blots

The cells were washed with PBS and lysed with CelLytic M (C2978, Sigma-Aldrich, Saint Louis, MO, USA) lysis buffer containing protease inhibitors. The lysates were centrifuged at 12,000 rmp for 5 min, and the supernatant was collected. The proteins were separated by Mini-PROTEAN^®^ TGXTM Precast Gels (BioRad, Berkeley, CA, USA) and transferred to a Trans-Blot Turbo Mini 0.2um PVDF Transfer Packs membrane (BioRad, Berkeley, CA, USA) by using Trans-Blot^®^ TurboTM Transfer System (Bio-Rad, Berkeley, CA, USA). The antibodies for *TOP2A* (HPA006458), *PLK1* (HPA053229), *MCM2* (HPA031496, a-tubulin (ab7291, Abcam) and GAPDH (sc47724, Santa Cruz Biotechnology, Inc., Dallas, TX, USA) were used for primary immunoblotting. All the antibodies were diluted at 1:1000 concentration. The membranes were incubated in primary antibody solution overnight at 4 °C with gentle rocking. Secondary antibody, goat Anti-Rabbit HRP (ab205718) or goat anti-mouse IgG-HRP (sc2005, Santa Cruz Biotechnology, Inc., Dallas, TX, USA), was blotted for 30 min at 4 °C with gentle rocking. The protein bands were detected with ImageQuant LAS 500 (29-0050-63, GE) automatic exposure procedure or 10 min exposure.

#### 2.9.4. Cell Viability Assay

Cell proliferation was detected by Cell Counting Kit-8 (CCK-8) assay. The 10 μL of CCK-8 reagent (1:10) was added to each well of 96-well plate with drug-treated cells as per manufacturer’s instruction. The 96-well plate was incubated at 37 °C for 2 h, and then the absorbance was measured at 450 nm using a microplate reader (Hidex Sense Beta Plus).

#### 2.9.5. Wound Healing Assay

The cells were seeded in a 6-well plate at 1 × 106 cells per well incubated at 37 °C until cells were 90–95% confluent. A scratch was created using a cell lifter, followed by washing with PBS to remove cell debris and then treated with MTX or WFA (4 uM) in a complete medium. After 24 h and 48 h of incubation, the cells were observed under a light microscope and photographed at 20× objective lens.

### 2.10. Molecular Docking Analysis

Molecular docking studies were carried out to rationalize the molecular basis and inhibitory potential of compounds MTX and WFA against TOP2A. In this connection, the crystal structure of TOP2A (PDB ID: 4FM9) with 2.9 Å resolution was downloaded from protein data bank [30]. The protein structure was processed by using the protein preparation module in MOE in order to fill the missing loops, atoms and assign bond orders [31]. Further minimization was performed using AMBER 10:EHT force field to handle the formal charges of the system. The 3D structure of the compound was built using a builder module in MOE. Energy minimization of the compounds was performed using MMFF94x force field [32]. MOE Site Finder package was used to predict the potential docking site on the protein surface with virtual atom probes. Next, 100 conformations were generated using induced fit protocol, while Triangle Matcher and London dG were used as placement and scoring method. The assessment of the docking results and investigation of the binding mode were carried out using MOE and Chimera [33].

## 3. Results

### 3.1. Survival Analysis Identifies Signature Prognostic Genes of HCC

We performed univariate Cox regression and KM analysis to investigate the correlation between the expression level of each protein-coding gene and patients’ survival outcomes in the LIHC and an independent Japanese cohort (LIRI-JP). We found that the expression level of genes is highly correlated between these two cohorts (r = 0.87; Figure 2A) and only included genes expressed in both cohorts for the survival analysis hereafter. Moreover, we checked the consistency of genes’ prognostic value between the two cohorts (Appendix A), which are 0.51 for Cox regression and 0.66 for KM analysis by Spearman correlation, respectively. As result, we identified 2999 and 7027 prognostic genes from Cox regression and KM analysis in the LIHC cohort, respectively, and 2949 of them are overlapped (*p* < 1.1 × 10^𢀒1016^, hypergeometric test). Similarly, in the LIRI-JP cohort, 1985 and 6141 prognostic genes were, respectively, identified in Cox regression and KM analysis, and 1881 genes were shared (*p* < 1.1 × 10^𢀒1016^, hypergeometric test). In addition, the consensus prognostic genes, which were identified as prognostic in both analyses of the LIHC and LIRI-JP cohorts were significantly overlapped (n = 1036, *p* < 1.1 × 10^𢀒1016^, hypergeometric test). Notably, most of the overlapped prognostic genes (n = 1004, 96.91%) were unfavorable prognostic genes, which means a high expression of these genes indicated poor patient survival outcome. As a result, the overlapped consensus prognostic genes (n = 1036) were defined as SPGs for indicating survival outcomes of HCC patients (Figure 2B). Functional GO analysis (Figure 2C) showed that unfavorable SPGs were involved in cell cycle process, DNA replication, cell proliferation and differentiation, posttranscriptional gene expression, which had been well acknowledged in cancer pathologies. Meanwhile, favorable SPGs were more enriched in various metabolic processes, including xenobiotic metabolism, immune process, etc., which are thought to be active anticancer metabolites in tumor cells [34].

### 3.2. Co-Expression Network Analysis Identifies Hub Modules of HCC

Next, we constructed GCN for both LIHC and LIRI-JP cohorts independently. In brief, we calculated the Spearman correlation coefficient between every gene pair based on their mRNA expression profiles and used only pairs with correlation ranked higher than 0.5% for the construction of the co-expression network. As a result, we obtained a GCN for the LIHC cohort with more than 600,000 gene–gene links, whose correlation’s 95% confidence interval ranged from 0.7094664 to 0.7096966. In addition, we identified 23 modules (Figure 3A) with more than 20 nodes and connectivity coefficients higher than 0.5. Similarly, we identified over 700, 000 edges with 95% confidence interval (0.6972966, 0.6975690) and 10 modules (Figure 3B) consisting of more than 20 nodes in the GCN of the LIRI-JP cohort.

Then, we superimposed the SPGs to all modules and found that five modules (M13, M22, M28, M57 and M80) were enriched with unfavorable SPGs, and one module (M23) was enriched with favorable SPGs in the LIHC GCN. In the LIRI-JP GCN, we only found two modules (M7 and M25) to be significantly overlapped with unfavorable SPGs. We also performed GO term enrichment analysis on these gene modules for the two GCNs to investigate their functional similarity (Appendix A). Interestingly, we observed most modules could be grouped into six parental biological processes and were conserved in both GCNs, as shown in Figure 3. For instance, several modules (M29 and M74), including the one enriched with favorable SPGs (M23), in LIHC GCN, were found to be related to cellular metabolic processes, including fatty acid oxidation, oxidative phosphorylation and fatty acid metabolic process, and similar metabolic processes, such as cellular glucuronidation, were also associated with modules M52 in the LIRI-JP GCN. This is consistent with our previous research [35], implicating a dysregulated lipid metabolism in the pathogenic and/or development process of HCC. Moreover, mitochondrial-related terms are enriched in modules M25 and M134 of LIHC GCN, as well as M77 in the LIRI-JP GCN. This could be explained by active mitochondria, which participate in many metabolic processes in HCC, including regulating oxidative phosphorylation, glucose metabolism and lipid metabolism [36]. In addition, we found DNA-replication-related biological processes were associated with unfavorable prognostic modules of both GCNs (M80 in LIHC GCN, M7 in LIRI-JP GCN) and M11, M61 in LIHC cohort, suggesting a dysregulated cell cycle regulation and an increased proliferation in the tumor, well-known hallmarks of cancer [37]. Moreover, we found several modules from both GCNs were related to RNA-related process. We found M13 and M28, which are enriched with unfavorable SPGs, as well as M30, M55, M83 and M122 in the LIHC GCN, were associated with RNA assembly, RNA metabolic processes and ribosome biogenesis, and M25, M40, M100, M255 in the LIRI-JP GCN were associated with mRNA processing, RNA export from nucleus and RNA metabolic process. Our analysis indicates that the transcription of the cell is elevated in cancer cells to boost enhanced proliferation together with the DNA replication process mentioned before. Immune response was found to be associated with M54 in LIHC GCN, as well as M12 in LIRI-JP GCN, indicating the activation of immune response in HCC. Interestingly, virus-related terms, such as viral gene expression, regulation of viral genome replication, were also enriched in modules from both GCNs, indicating potential virus-associated HCC cases in both cohorts [38]. Finally, modules found with no clear function category were grouped as Others, responsible for regulating cellular functions and organism development.

### 3.3. Identification of Target Genes in HCC

To identify consensus prognostic gene modules between the two GCNs, we compared all prognostic modules in LIHC GCN (M13, M22, M23, M28, M57 and M80) and LIRI-JP GCN (M7 and M25) in a pairwise way (Appendix A) and identified two significant overlapped (FDR < 0.5) module pairs: M80 in LIHC cohort and M7 in LIRI-JP cohort, M57 in LIHC cohort and M25 in LIRI-JP cohort. Notably, M80 (n = 674) in LIHC cohort and M7 (n = 1183) in LIRI-JP cohort were significantly overlapped (n = 401, FDR = 8.018762 × 10-222, adjusted by BH, hypergeometric test). In addition, 314 of the 401 overlapped genes (78.3%) between these two prognostic modules were SPGs (Figure 4A). We also performed topology analysis on M80 of LIHC GCN and M7 of LIRI-JP GCN to extract hub genes from these two modules. Interestingly, we found the degree, betweenness and closeness of the 401 overlapped genes were highly correlated (r = 0.86 for degree, r = 0.65 for betweenness and r = 0.82 for closeness, Appendix A), implying similar network structures of M80 and M7, regardless of the heterogeneity between these two cohorts. Besides, M57 (n = 39) in LIHC cohort and M25 (n = 27) in LIRI-JP cohort shared four overlapped genes (FDR = 5.24 × 10^−05^, adjusted by BH, hypergeometric test). Two of the overlapped genes were also found as SPGs.

Next, we sorted all genes in the overlapped module pairs according to the descending order of topological parameters and compared the top 10% genes in both modules based on their topology. As a result, 67 top genes from M80 and 118 genes from M7 were extracted according to their corresponding module sizes. Interestingly, the two top-degree gene subsets obtained from M80 and M7 were significantly overlapped (n = 47, *p* < 0.05, hypergeometric test), and all these overlapping genes were also SPGs. Likewise, we found 16 genes (FDR < 0.05) with top betweenness values and 39 genes (*p* < 0.05) with top closeness values shared by two modules, and the majority of them were SPGs too (n = 15, 93.75% for betweenness, n = 38, 97.43% for closeness). To identify the hub gene shared by M80 and M7, we merged the top-ranked genes, which were overlapped with SPGs (n = 47 for degree, n = 15 for betweenness and n = 38 for closeness) and extracted 57 unique genes as common hub genes from the M80–M7 module pair. We did not identify common hub genes in the M57–M25 module pair. Therefore, the common hub genes in the M80–M7 pair were selected for further analysis, as their high expression not only possesses central influence over the two modules, but it also significantly contributes to patient survival outcome.

Moreover, we examined the essential score of the hub genes in 16 primary HCC cell lines obtained from the DepMap data portal. The essential scores were estimated by the genome-scale CRISPR-Cas9 viability screens in various cell lines to identify specific genetic dependencies, and lower essential scores indicated higher influence on tumor cell proliferation and DNA replication. In this context, we selected 10 genes (*RAN*, *PLK1*, *KIF11*, *CDK1*, *TOP2A*, *CHAF1B*, *KIF23*, *POLD1*, *MCM2*, *GINS1*) with essential score of less than −1.5 as potential target genes (Figure 4B). As we expected, these hub genes have very similar topological attributes in M80 and M7 (Appendix A) and show consistent unfavorable association with survival of HCC patients (Appendix A).

We also investigated the expression levels of the 10 hub genes based on their mRNA expression profiles of LIHC normal and tumor tissues. As shown in Figure 4C, all hub genes showed higher expression in tumor tissue compared to the adjacent normal tissue. Most of these target genes (except for *RAN*, *POLD1* and *MCM2*) presented very low average expression level (TPM < 1) in normal tissues, and their expression levels were significantly increased in the liver tumor tissues. To evaluate their expression at protein level, we also examined the IHC staining images of the target genes (Figure 4D) in normal and tumor tissues from the HPA [28]. Nine of the ten target genes with selected antibodies were tested in HPA (except for *RAN*), and these nine genes showed higher protein expression in tumor cells than corresponding normal cells. For example, *TOP2A* (with antibody HPA006458) was not detected in normal hepatocytes cell but showed high and intensive staining in tumor cells. As these target genes encode cell-cycle-related proteins and are involved in cell division processes, this alteration may indicate elevated tumorigenesis or tumor progression.

Besides the expression level differences of target genes in normal and tumor tissues, great differences were also observed between target genes. We further performed scRNA-seq analysis to examine the expression of target genes at cell level (Appendix A). Compared to others, *RAN* was found expressed in almost all cell types, which can explain the high expression level compared to other candidate target genes. We also observed most genes were expressed specifically in three cell types: hepatocytes, T cells and Kupffer cells, indicating the essential roles of macrophage and immune processes in HCC pathology.

### 3.4. Drug Repositioning for HCC

To locate the potential drugs for candidate target genes, we performed a profile-based drug repositioning by integrating the transcriptional response of shRNA knockdown and chemical perturbation from the CMap (https://clue.io/, version: CMAP LINCS 2020, 22 June 2021), as shown in Figure 5A.

To investigate the effective drugs for HCC, we performed drug repositioning analysis following the flowchart in Figure 5A. The steps were conducted as follows: (a,b) shRNA and drug screening for target genes. We mapped seven target genes into the shRNA (matrix a)- and drug (matrix b)-gene pairwise association matrices of HepG2 cell line, in which the moderated z-scores represent the averaged effects of three biological replicate perturbations. For every drug, one gene was treated by multiple-dose portions and time points. Notably, three genes (*KIF23*, *PLOD1* and *GINS1*) were excluded from this analysis because their shRNA knockdown information was not available in CMap. In total, 30 shRNAs among seven target genes (3 shRNAs for *RAN*, *PLK1*, *KIF11*, *CHAF1B*, *MCM2*, separately, 6 shRNAs for *CDK1* and 9 shRNAs for *TOP2A*) were extracted in the drug–gene and shRNA–gene interactive matrix; (c) Construction of drug–shRNA matrix. Drug–shRNA matrix between seven target gene knockout shRNAs and drug compounds was constructed by integrating the drug–gene and shRNA–gene interactive matrix, consisting of 30 columns (shRNAs) and 18,157 rows (drugs). The matrix coefficients calculated by Spearman correlation represent the similarity of effects on gene expression induced by specific shRNA knockout and drug compound treatment. (d) Optimize the drug–gene correlation matrix. Among all the drug–shRNA pairs, we prioritized the correlation matrix by keeping the strongest dose- and time-point-responsive connections for every drug, since partial drugs were tested on shRNAs with multiple processing doses and times. The clustering analysis was also performed for the optimization of the shRNAs. Only the cluster with higher median correlation coefficients (marked as MdnCorr in Figure 5A) was reserved for the following analysis (Appendix A). As a result, 18 shRNAs and 5284 drugs were kept for later analysis; (e) Rank drugs by correlation coefficient. We ranked all related drugs with positive order of correlation coefficient and marked the drugs with their rank number. Since the target genes also possess multiple shRNAs, we only kept the common drugs with top 5% (n = 264) correlation rank with every shRNA corresponding to the target gene to improve the accuracy of our approach (Appendix A); (f) Selection of three most effective drugs for each target gene. For every target gene, the corresponding drug compounds were marked by the median rank of the reserved shRNA ranks, in which higher median rank means that the drug has higher overall interactive intensity to the target gene. Proceeding from the whole process, we kept shared drugs between all reserved shRNAs and selected the top three drugs as the most potentially effective drugs for the target gene (Figure 5B, Appendix A). In particular, WFA was proposed as a potential drug for two target genes, *KIF11* and *TOP2A*, and sulforaphane was predicted to be effective for both target genes *PLK1* and *MCM2*.

We performed literature and clinical trial review for potential drugs we identified with correlation above 0.4 (Figure 5B), which included the drugs identified for inhibition of *TOP2A*, *PLK1* and *MCM2*, and finally validated their effects on HepG2 cell line. Linsitinib is the top candidate identified for targeting *PLK1*, and a phase II trial with linsitinib (NCT01334710) was initiated but subsequently terminated for safety reasons. A clinical case reported crizotinib, which is predicted to be able to inhibit *MCM2* based on our analysis, could lead to fatal liver injury or failure [39]. Thus, linsitinib and crizotinib were excluded from our experiment. All the other available drugs were purchased from Selleckchem (https://www.selleckchem.com/, purchased at 13 September 2021). To evaluate drug effectiveness, we treated HepG2 cells with drugs for 24 h and isolated the protein. Then, we performed Western blotting to detect the changes in protein expression of *TOP2A*, *PLK1* and *MCM2*. WFA and MTX strongly suppressed *TOP2A* expression in the cells (Figure 6A). In contrast, AZD-6482 increased *TOP2A* expression unexpectedly (Figure 6A). The expression of *PLK1* or *MCM2* was not significantly affected by the identified drug candidates (Figure 6A). The results of the proliferation assay conducted after 24 h treatment showed that MTX and WFA significantly inhibited the proliferation of HepG2 cells (Figure 6B). The intensity ratio and other detailed results can be found in Appendix A.

Next, we validated the effect of MTX and WFA on migration of HepG2 cells by scratch wound-healing assay. As shown, the migration of HepG2 cells was attenuated by MTX and WFA after 24 h and 48 h treatment (Figure 6C). In addition, we observed that HepG2 cells were mostly dead from WFA after 48 h treatment (Figure 6C). Our results suggested MTX and WFA could target the proposed target gene *TOP2A* and could, thus, be promising drugs for HCC treatment.

Molecular docking studies were performed to comprehend the plausible ligand binding mechanism of MTX and WFA with *TOP2A*. The proposed binding mode of MTX and WFA showed good binding affinity of −8.214 and −7.9625, respectively. The interacting residues of *TOP2A* (generated by Chimera) with both compounds are demonstrated in Figure 6D,E. As evident from the figure, MTX presented very good binding pattern with *TOP2A* by showing numerous hydrophilic and hydrophobic interactions. Hydrogen bonding plays a significant role in the binding affinity in which Asn487 (2.8 Å), Asn504 (2.1Å) and Glu506 (1.8Å) residues are involved. Hydrophobic interactions are also formed by Lys489, Ile490, Met762, Ser763 and Asp831. Besides these interactions, the polycyclic aromatic core of MTX is also involved in pi-stacking interaction, which facilitates the intercalation into the DNA cleavage site.

The top-ranked docked pose of the WFA is presented in Figure 6E. As evident from the figure, the steroidal part of WFA was stabilized by hydrophobic interactions with the surrounding residues Lys489, Ile490, Asp543 and Ser763, while a hydrogen bond interaction was observed between the hydroxyl group of WFA with the main chain of Lys489 at a distance of 2.5 Å. The compound exhibited additional salt bridge interaction within the binding site with Arg487. Further, ligand protein interactions, stabilized by arene-H interactions with the DNA nucleotides, might help in enzyme-mediated DNA cleavage.

## 4. Discussion

HCC is a common malignancy of the liver with increasing prevalence and poor 5-year survival rate worldwide. To improve the efficiency of treatment development for HCC, continued efforts are needed to identify novel drug targets and drugs for effective HCC treatment strategies. It is well acknowledged that drug repositioning has been the safe and economic way to explore new drug treatments for different kinds of diseases. In this study, we identified therapeutic targets and effective drugs for treating HCC based on the combination of GCN analysis and gene perturbation expression profiles from heterogeneous data sources. Drug effectiveness was also validated by performing in vitro experiments, implying a promising application of our drug repurposing approach for other cancer types.

In this study, we extracted 1036 prognostic genes among HCC patients from two independent cohorts, with the majority of genes demonstrating unfavorable association with patient survival outcome. In addition to the uncertain liver cancer subtypes in the Japanese cohort, and geographical differences between these two cohorts, the SPGs still showed significant overlap between these two cohorts. In previous research, we identified a general functional pattern among prognostic genes in 17 major cancer types [21], where the unfavorable genes were reported to be associated with cell proliferation and dedifferentiation, leading to poorer patient survival. The result of functional analysis of HCC SPGs accorded with the cancer pattern and further identified specific prognostic genes in HCC patients among two independent cohorts.

We constructed GCN for each HCC cohorts and identified high-transitivity modules based on network components and structure. The functional analysis of modules demonstrated the consistent pathology among different patient cohorts. One well-known main risk factor of HCC is virus infection, including HBV and HCV infection, and it accounts for more than 70% of all HCC incidence worldwide [40]. In our study, virus-related modules were identified in both cohorts, indicating the possibility of high proportion of virus-induced HCC cases in the present study. Moreover, the modules involved in fatty acid metabolism were identified and occupied a great portion in both cohorts; these modules have been considered as NAFLD-associated modules and related highly to disease progression toward HCC [35].

This network-based analysis also enabled the identification of key genes associated with the development of HCC. All ten target genes we identified were not only hub genes with high topology ranking but also prognostic genes associated with clinical living conditions, which means the perturbation of these driver genes in modules creates systematic effects among GCN and survival outcomes. Three target genes, *RAN*, *PLK1*, *TOP2A*, had been proven to have synergistic effects in the development of HCC [41,42,43]. The *KIF11*, *CDK1*, *MCM2* and *GINS1* genes were also identified as potential targets for HCC by independent analysis or experiments [44,45,46,47,48]. Although *CHAF1B*, *KIF23* and *POLD1* were not mapped into CMap, previous studies reported their potential therapeutic effect using HCC cell line models or animal models [49,50,51]. The expression of these ten target genes was significantly upregulated in the tumor tissues compared to the adjacent normal tissues (Figure 4D), and three of them (*PLK1*, *CDK1*, *TOP2A*) were also marked as landmark genes in CMap, which means these genes were widely expressed across the lineage. Thus, it suggested that the drugs targeting these landmark genes are potentially broad-spectrum anti-cancer drugs.

After considering both the effect of potential drugs and shRNA perturbation physiology in HepG2 cell line, we identified and validated the top effective drugs with the highest impact on target genes. MTX and WFA, as candidate HCC drugs targeting *TOP2A*, showed great efficiency in suppressing *TOP2A* protein expression, as well as HCC cell proliferation in HepG2 cells. Both of them were reported to inhibit autophagy in HCC cells [52,53,54,55]. Autophagy inhibition enhanced cell death in HCC cell lines [56] and showed a markable suppression of HCC in both in vivo and in vitro studies [57]. Notably, a phase II clinical trial showed MTX can promote HCC patients’ survival with limited side effects [58]. A considerable correlation of mRNA expression profiles between cell line and primary tissues in liver cancer was observed [54], indicating the promising clinical application for the effective drug we identified. Further application of our method is currently in progress in our laboratory.

## 5. Conclusions

In conclusion, we proposed an mRNA expression profile-based drug repositioning method to repurpose existing drugs to HCC. This GCN-dependent method enabled the identification of genes, which play key roles in the survival of patients and possess essential functions in the co-expression network based on the network topology. The comprehensive assessment of drug-perturbation effects on genes allowed us to retrieve the most effective drugs for target genes in HCC with in vitro evidence for drug effectiveness on tumor cells, highlighting the promising future for our approach. In this context, more cancer types should be considered with our GCN-based method for exploring the potential drugs with less cost and higher efficacy.

## Figures and Tables

**Figure 1 cancers-14-01573-f001:**
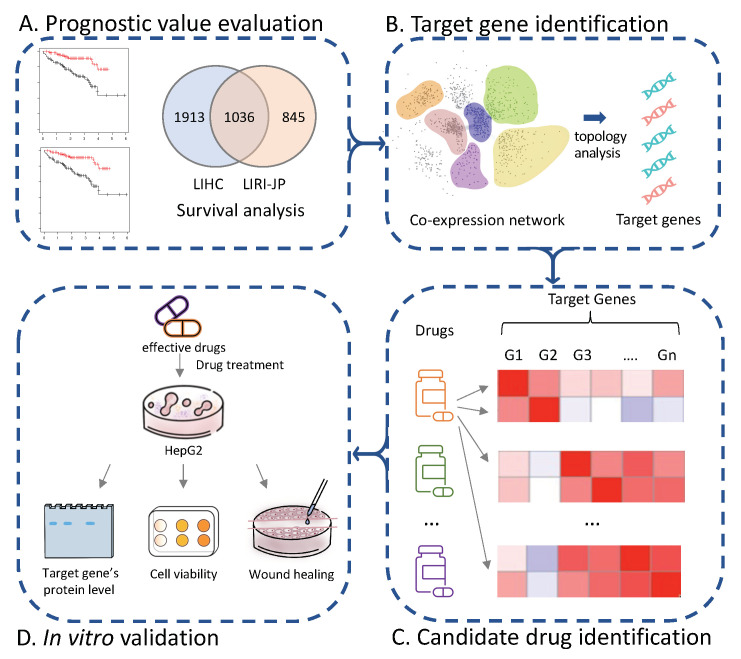
Flow chart of systematic drug repositioning approach for HCC.

**Figure 2 cancers-14-01573-f002:**
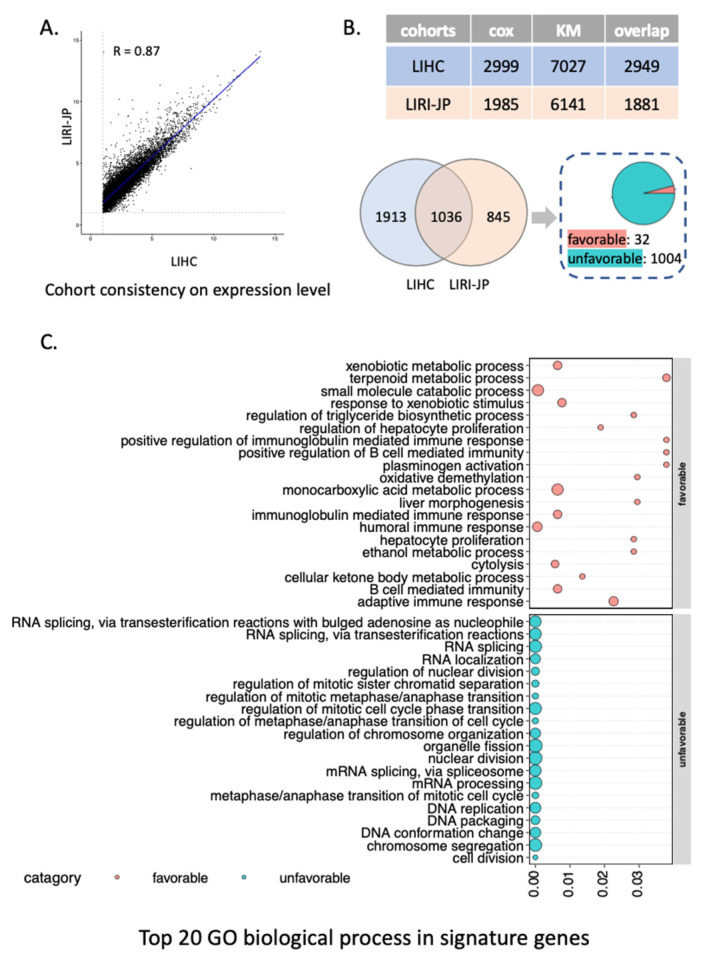
Identification and functional analysis of HCC SPGs. (**A**) The correlation plot shows great consistency in gene expression level between LIHC and LIRI-JP cohort. (**B**) Identification of signature prognostic genes in LIHC (marked with blue color) and LIRI-JP cohorts (marked with orange color). The table shows the number of prognostic genes in Cox survival analysis and KM analysis in two cohorts. We further identified the 1036 SPGs shared by both prognostic gene sets (Venn diagram). (**C**) Functional analysis showed the top 20 most significantly GO terms in favorable and unfavorable SPGs, presented with pink and green dots, respectively.

**Figure 3 cancers-14-01573-f003:**
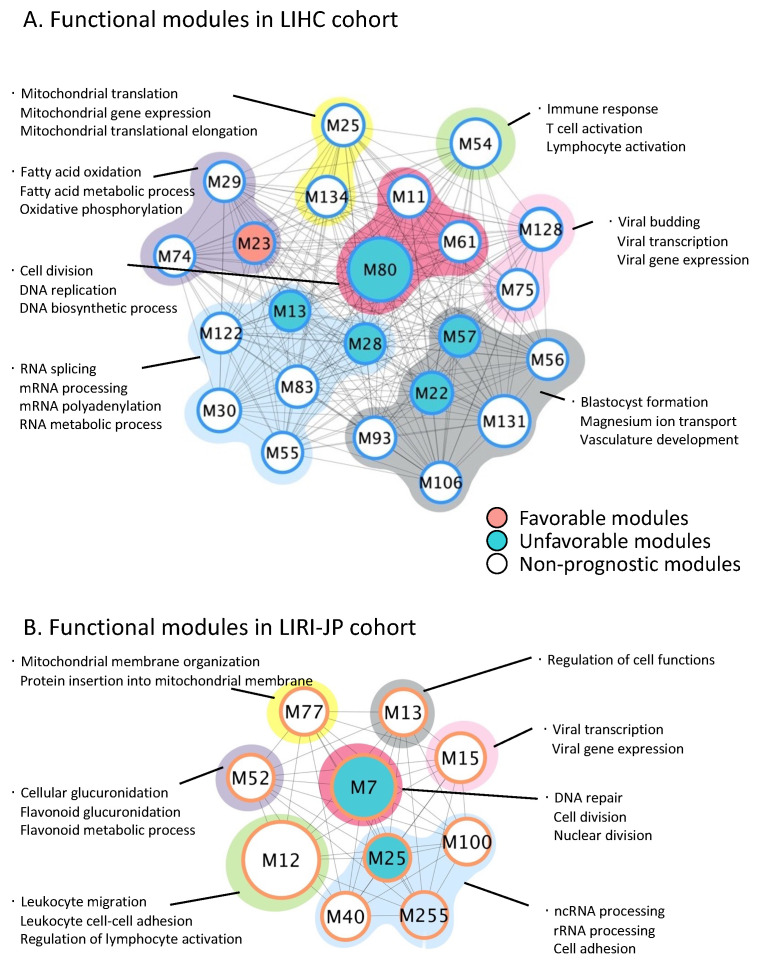
High-centrality functional modules in HCC cohorts. The networks were limited to modules with a minimum number of 20 nodes and a connectivity coefficient larger than 0.5. (**A**) and (**B**) showed the modules identified in LIHC and LIRI-JP cohort, respectively. The prognostic attributes of modules were marked by different color, as shown in legend. Modules with similar biological functions were circled with same background color (red—DNA replication, green—Immune response, purple—Metabolic process, yellow—Mitochondrial process, blue—RNA-related process, pink—Virus infection and gray—Other functions). Top biological processes were listed beside the functional circle for detailed information.

**Figure 4 cancers-14-01573-f004:**
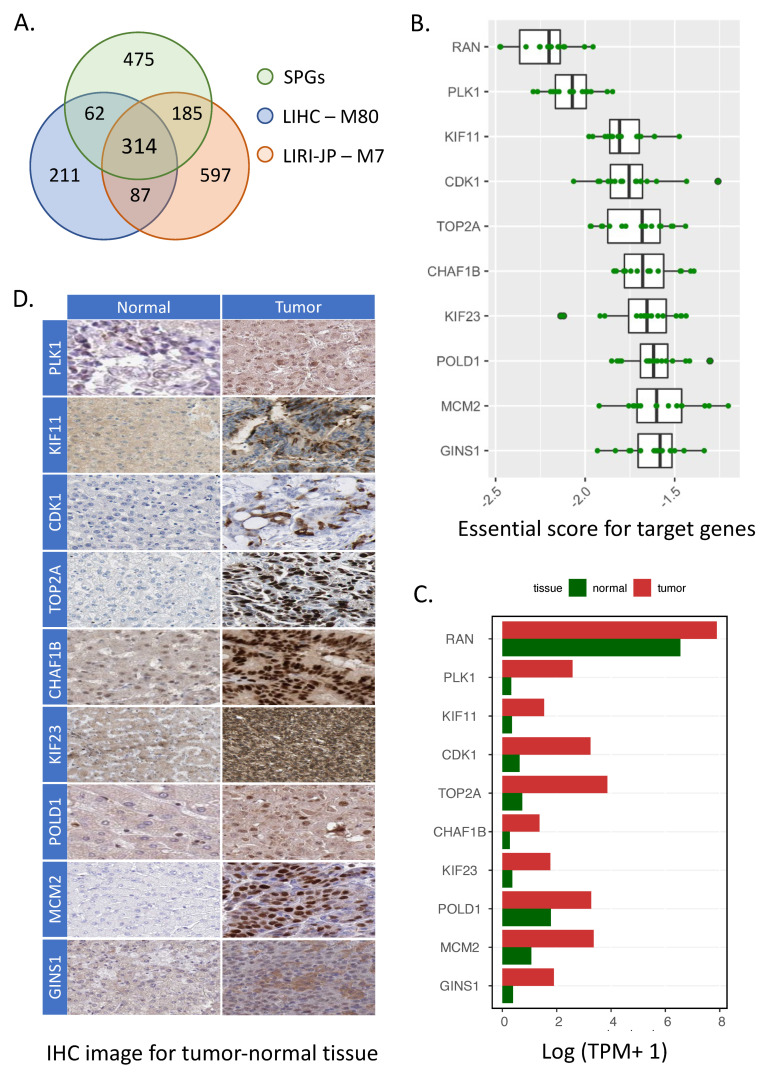
Identification of HCC target genes. (**A**) Venn diagram showed the relative overlapping outcomes of M80 (LIHC module), M7 (LIRI-JP module) and SPGs. (**B**) Essential scores for potential target genes in 16 primary HCC cell lines. (**C**) Protein-staining IHC images for potential target genes among normal and liver tumor cells. (**D**) The average gene expression level of target genes in normal and tumor tissues among 50 HCC patients.

**Figure 5 cancers-14-01573-f005:**
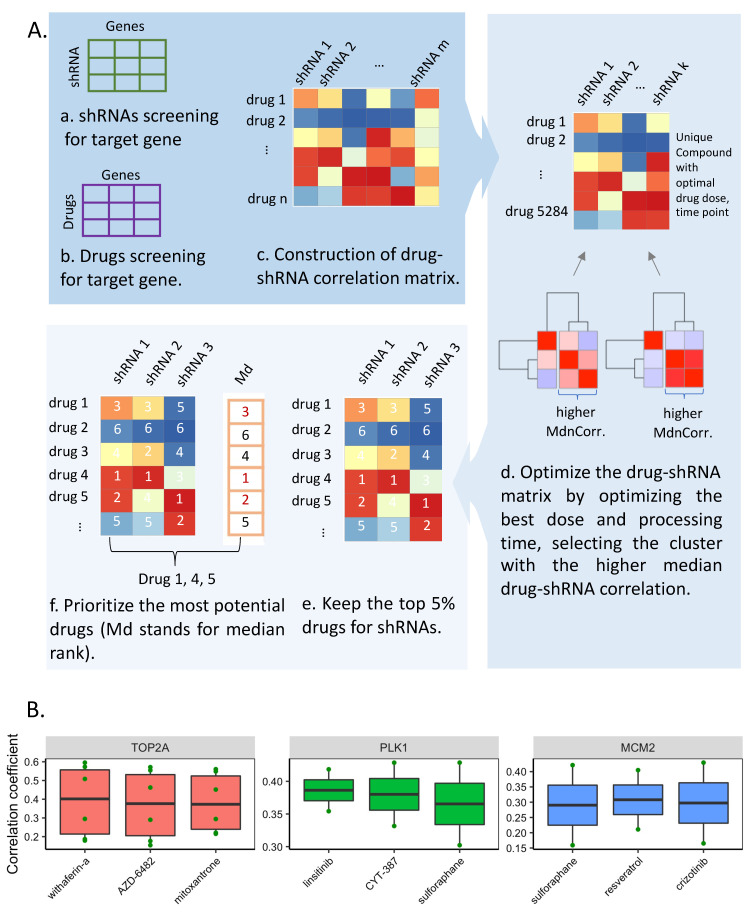
Drug prediction for HCC target genes. (**A**) Workflow of drug identification for target genes. The MdnCorr here stands for the median correlation coefficient. (**B**) The box plot showed top three effective drugs for each target gene. Each point in the box plot represents a shRNA for knockdown of corresponding target genes.

**Figure 6 cancers-14-01573-f006:**
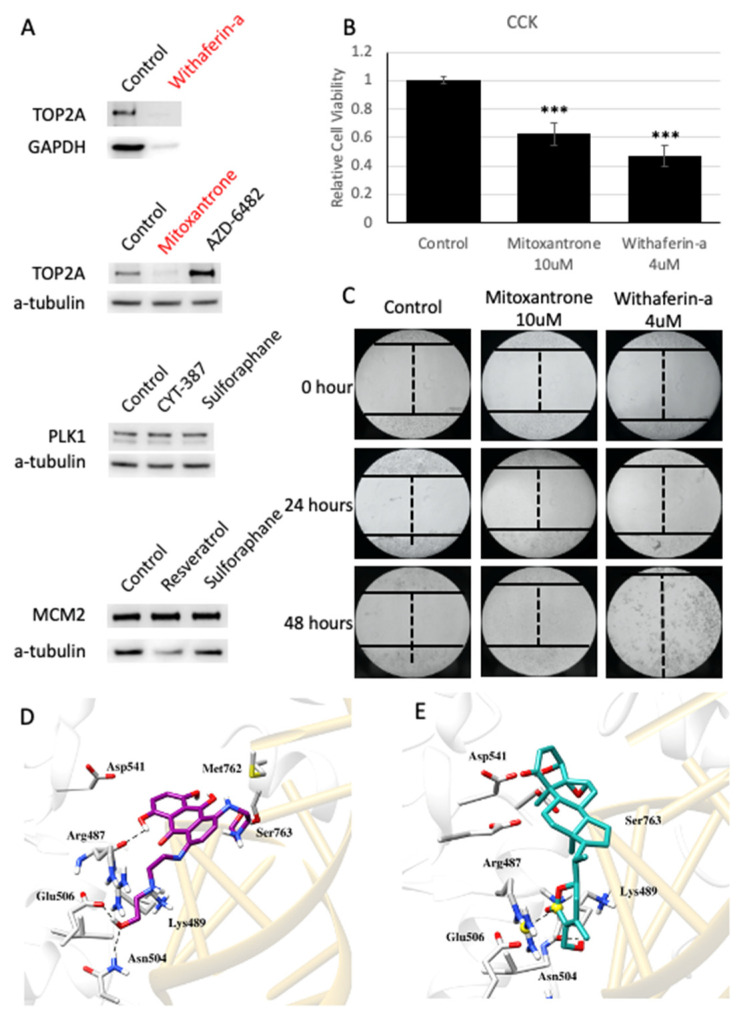
Validation of top effective drugs. (**A**) Protein expression changes with drugs treatment in *TOP2A*, *PLK1* and *MCM2*. (**B**) The proliferation assay showed MTX and WFA significantly suppressed *TOP2A* formation in HepG2 cell line (*** means *p* < 0.001 in *t*-test). (**C**) The scratch wound-healing assay showed MTX and WFA strongly inhibit HepG2 cells migration. (**D**) The docked conformation of the MTX inside the binding site of *TOP2A*. H-bond interactions were represented as black dotted lines. (**E**) The docked conformation of the WFA inside the binding site of *TOP2A*. H-bond interactions were represented as black dotted lines.

## Data Availability

The RNA-seq and scRNA-seq data we used in this study were downloaded from public data resources. We have clarified all the accession code in the Materials and Methods section.

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
