# Peer review of "A Gene Co-Expression Network-Based Drug Repositioning Approach Identifies Candidates for Treatment of Hepatocellular Carcinoma"

_cancers, 2022, doi:10.3390/cancers14061573_

Round 1
Reviewer 1 Report
The content of this manuscript is relatively interesting, and there are few scientific problems.
However, there is a fatal problem with this submission.
First, I am missing the necessary information that is worth posting because we do not follow the posting rules. You must use the template to correct it to the appropriate format and fill in the missing information.
The second, file submitted does not have any supplementary material and cannot be judged.
Based on these problems, I consider that this manuscript is in a manuscript that has not reached the submission stage and considers it as "reject".
You must recreate it from scratch and officially resubmitted it before considering whether it needs to be revised here.
Reviewer 2 Report
The authors applied a gene co-expression network analysis to discover target genes for drug repositioning in hepatocellular carcinoma (HCC), including originally developed drug repositioning method using shRNA, drug perturbation of liver cancer cell lines, and in-vitro evaluation. The work is overall systematic and sound. The reviewer has some minor comments.
General comment 1: Two words seem to have been used for the same meaning, LIRI-JP cohort and ICGC cohort. Please use only one word throughout the manuscript to avoid confusion.
Page 2, the first paragraph: Please be careful with the term “with different races” at the end of the paragraph, as it is difficult to define them for US. Perhaps, ‘geographic location’ can be a better term in this case.
Page 2, the third paragraph: Please consider providing additional descriptions or examples of systems biology approaches for identifying target genes.
Page 10, line 216: Please provide the version of Cancer Dependency Map (DepMap) data portal used in this study because the database is updated quarterly.
Page 10, line 231: Please add an elaboration why the nine genes were selected out of 10 hub genes for evaluating their protein expression levels.
Page 11: Description for figure 5.A.b is absent in the Results section.
Page 15, Materials and Methods 4.1: For the gene expression data from TCGA, it is unclear whether the FPKM data of samples or raw count data calculated by Kallisto were used in this study. I understand that Kallisto was used to sort protein-coding genes out of whole RNA-seq data based on the GENCODE. Additional explanation looks necessary beyond “quantified the mRNA expression by Kallisto”.
Page 17, Material and Methods 4.5: Please provide a reference for the criteria, -0.05, in the sentence, “In most cases, gene with essential score less than -0.05 represents the cell will grow slower when the gene is knocked out”.
Figures 1, 5: Please consider re-organizing the direction of arrows that are presented in multiple directions.
Figure 2A: A process for preparing the correlation plot between two cohorts need to be provided in the Material and Methods section.
Figure 5: Please provide the definition (or full name) of MdnCorr.
Minor comments
Please pay attention to the language throughout the manuscript. Following are a few examples.
Page 2, line 48: “Continuous” -> “continuously”.
Page 2, line 51: “depends” -> “depend”
Page 3, line 96: “LINSC” -> “LINCS”.
Page 7, line 149: “super imposed” -> “superimposed”
Page 10, line 236: The sentence, “Besides the expression level differences of target genes in normal and tumor tissues, great differences were also observed between target genes.”, can be located in the next paragraph about scRNA-seq.
Page 12, line 262: “incl ude” -> “includes”
Page 14, line 319: “significantly” -> “significant”
Page 15, line 363: “GCN based” -> “GCN-based”
Round 2
Reviewer 1 Report
The manuscript has been improved as requested.